# The Effect of Short Chain Carboxylic Acids as Additives on the Crystallization of Methylammonium Lead Triiodide (MAPI)

Chiara Dionigi [1,*], Meriem Goudjil [1,2,*], Giampiero Ruani [1] and Luca Bindi [2]

1   Consiglio Nazionale delle Ricerche, Istituto per lo Studio dei Materiali Nanostrutturati (ISMN), Via P. Gobetti 101, I-40129 Bologna, Italy
2   Dipartimento di Scienze della Terra, Università degli Studi di Firenze, Via La Pira 4, I-50121 Firenze, Italy
*   Correspondence: chiara.dionigi@cnr.it (C.D.); meriem.goudjil@unifi.it (M.G.)

**Abstract:** Due to their exceptional properties, the study of hybrid perovskite (HyP) structures and applications dominate current photovoltaic prospects. Methylammonium lead tri-iodide perovskite (MAPI) is the model compound of the HyP class of materials that, in a few years, achieved, in photovoltaics, a power conversion efficiency of 25%. The attention on HyP has recently moved to large single crystals as emerging candidates for photovoltaic application because of their improved stability and optoelectronic properties compared to polycrystalline films. To control the quality and symmetry of the large MAPI single crystals, we proposed an original method that consisted of adding short-chain carboxylic acids to the inverse temperature crystallization (ICT) of MAPI in γ-butyrolactone (GBL). The crystals were characterized by single-crystal X-ray diffraction (SC-XRD), X-ray powder diffraction (XRPD) and Raman spectroscopy. Based on SC-XRD analysis, MAPI crystals grown using acetic and trifluoroacetic acids adopt a tetragonal symmetry "*I4cm*". MAPI grown in the presence of formic acid turned out to crystallize in the orthorhombic "*Fmmm*" space group demonstrating the acid's effect on the crystallization of MAPI.

**Keywords:** hybrid perovskite; methylammonium lead tri-iodide; single-crystal; inverse temperature crystallization; short chain acids

## 1. Introduction

Hybrid perovskites (HyP), in a few years, have led to obtaining cells with very high efficiency using relatively simple and low energy consumption manufacturing technologies. Nevertheless, despite the superior qualities of HyP, the crucial issue in HyP research and exploitation remains enhancing their performance and stability [1–4].

The 3D structure of HyP has the formula $ABX_3$, where A is an organic or alkali metal cation, B is a divalent cation (typically $Pb^{2+}$, $Sn^{2+}$, or $Ge^{2+}$), and X is a halogen. This structure, in particular for lead halides, is low modulable since the tolerance factor is satisfied only for caesium, methylammonium ($MA^+$), hydroxylammonium ($NH_3OH^+$), hydrazinium ($NH_2NH_3^+$), azetidinium (($CH_2)_3NH_2^+$), formamidinium ($FA^+$), and imidazolium ($C_3N_2H_5^+$) cations [5–7].

These HyPs, in particular, are of interest due to their tuneable optoelectronic properties, including quadratic non-linear optical (NLO) properties (in the case of methylhydrazinium analogues) [8]. Moreover, the iodides, especially $MAPbI_3$, are of great interest due to their superior photovoltaic properties [1–4].

HyP single crystals contain six orders of magnitude fewer traps with respect to the trap densities of their polycrystalline films [9–12]. In force of this evidence, the HyP mono-crystal is considered the model to aim for applications in photovoltaics. Consequently, attention has recently moved to single perovskite crystals as emerging candidates for photovoltaic applications [9].

For these reasons, the research on HyP crystallization is currently vigorously active [10], and several methods, such as the anti-solvent method [11] and the in situ solvent conversion method [12], were applied to grow high-quality HyP single crystals.

An efficient and quick method, based on inverse temperature crystallization (ITC), was also proposed in the last five years to synthesize $MAPbX_3$ (X: Cl, Br and I) large crystals. This method was based on the loss of perovskite solubility with the temperature increase in a specific solvent. The choice of solvent has always been a crucial point for the quality of the resulting crystals [13–15]. In particular, in γ-butyrolactone (GBL), dimethyl-sulfoxide (DMSO), or dimethyl-formamide (DMF), raising the temperature to 120 °C induces the nucleation and growth of hybrid perovskite, although the high temperature inevitably leads to the deterioration of the crystal quality, especially at the surface, due to the volatility of the organic cation [13,16].

The crystallization of methylammonium lead triiodide ($MAPbI_3$), hereafter called MAPI, by the ITC method is possible only in GBL as a solvent [13].

Nayak et al. previously demonstrated that a rise in acidity due to GBL ageing increases the concentration of the reagents and decreases the solvent's strength. In this way, the supersaturation and the consequent crystallization were achieved at a lower temperature [17].

Zhu et al. [18] recently demonstrated that acid as an additive allows the crystallization of larger perovskite grain size with reduced defect density through the Ostwald ripening mechanism. Indeed, the acid induced the small particle dissolution to grow the large particles even larger. In addition, they demonstrated that the organic fluorinated groups on the acid additive increased the perovskite hydrophobicity, whose films could resist even high humidity for long-term stability.

Similarly, Meng et al. [19] investigated the effect of formic acid on the size of the colloids in the precursor solution of formamidinium iodide (FAI) and lead iodide ($PbI_2$). They demonstrated that formate anions have the role of passivating agents for $FAPbI_3$ perovskites improving the efficiency of the perovskite solar cells.

Due to this stimulating research on HyP [17–19], we selected short acids as additives in the ICT crystallization of MAPI.

In particular, we chose formic acid, acetic acid and trifluoroacetic acid, which are similar short-chain acids in structure and dimensions but with significantly different acidities.

We investigated, by single-crystal X-ray diffraction, the structural features of MAPI crystals generated by the ITC method in the presence of the selected acid additives. Moreover, the synthesized MAPI compounds were assessed by means of X-ray powder diffraction and Raman measurements.

Several crystal structures were reported for MAPI [20–23]. In particular, MAPI phase transitions were investigated by Mark Weller et al. [24], who reported that MAPI has three structural modifications: a cubic *Pm-3m* phase above 327 K, a tetragonal *I4/mcm* phase between 165 to 327 K, and an orthorhombic *Pnma* phase below 165 K. Within a temperature range, the coexistence of the two phases cannot be excluded.

Since the electronic properties of MAPI strongly depend on the structural characteristics of the MAPI phase, as reported by Zhong et al. [25], crystallographic characterization has become extremely important as a future prospective interest in the acid effect on MAPI optimization.

## 2. Results

### 2.1. Crystal Habit

Figure 1 reports the optical microscope images of the MAPI crystals obtained with acid additives in the GBL, as reported in the Materials and Methods section.

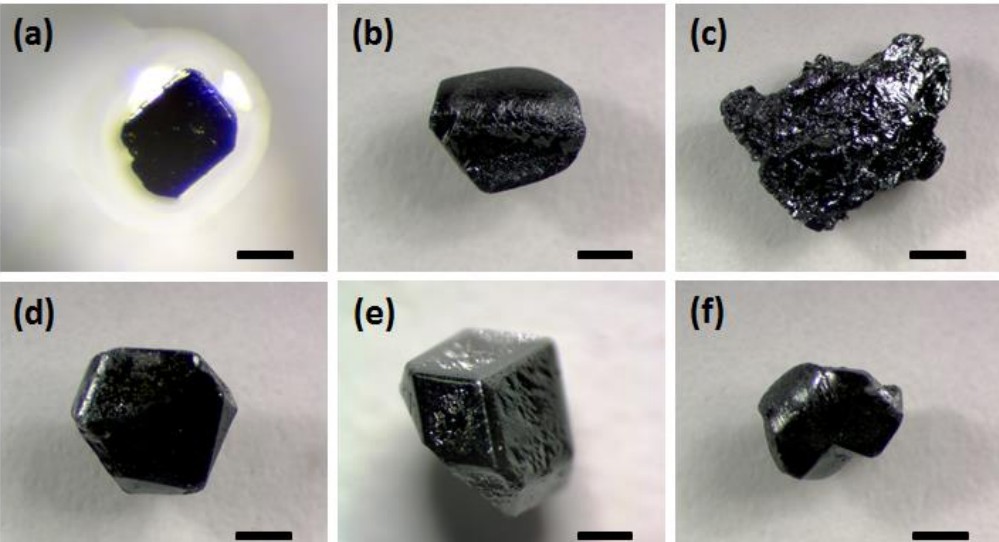

**Figure 1.** Optical microscope images (scale bar: 1 mm) of the hybrid perovskite crystals grown in γ-butyrolactone (GBL) with acid additives: (**a**) Crystal O, without additive, (**b**) crystal A nucleated and grown with acetic acid, (**c**) crystal B nucleated and grown with formic acid, (**d,e**) crystals B$_{seed}$ nucleated with of acetic acid and grown in the presence of formic acid, and (**f**) crystal C nucleated and grown with trifluoroacetic acid.

The crystals were named O (no acid additive), A (acetic acid as additive), B (using thrice concentrated with formic acid as an additive), B$_{seed}$ (nucleation with acetic acid and growth with formic acid as an additive), and C (trifluoracetic acid as an additive).

All the grown crystals are black in appearance and exhibit a millimetric size with dimensions of >2 mm, with the exception of crystal O (Figure 1a), which has a thickness of ≤0.3 mm. They display different forms: crystal O adopts a tabular morphology (Figure 1a), crystals A and C grew as large blocks (Figure 1b,f), crystal B, using thrice concentrated formic acid, has an undefined shape (Figure 1c), and finally crystal B$_{seed}$, nucleated in the acetic acid and then grown in formic acid, formed as dodecahedra with hexahedral-like (Figure 1d) and rhombohedral-like (Figure 1e) facets.

## 2.2. Crystallographic Study

The crystal structures were solved by the intrinsic phasing method [26] and refined by the full-matrix least-squares minimization based on $F^2$ using the *SHELX*L program [27] implemented with Olex2 software [28].

The structure solution of all the MAPbI$_3$ samples was conducted as follows: The unit-cell parameters were chosen on the basis of the reflection indexing percentage; the space group assignment was undertaken according to the reflection conditions (improved by $R_{int}$ and $R_\sigma$ values) and taking into account the perovskite symmetry hierarchy of the aristotype, cubic *Pm*-3*m*; and, finally, on the basis of successful structural refinement. Heavy atoms, Pb and I, were located first in the Fourier maps, and C and N atoms were subsequently located in the refinement cycles. Neutral scattering curves, taken from the International Tables for Crystallography [29], were used for all the atoms. In some cases, some constraints were applied to refine the C and N disorder. Pb, I, N and C atoms were all anisotropically refined. The assignment of C and N coordinates was consistent with the methylammonium (MA)$^+$ geometry inside the cubo-octahedral cavities formed by I atoms.

Although X-ray diffraction is a very useful technique and represents the primary source of crystallographic information for the hybrid perovskites, structural analysis remains partial for HyPs. The identification of carbon, nitrogen and hydrogen positions cannot be unambiguously determined by means of X-ray diffraction. In fact, the scattering is dominated by the higher atomic number elements (i.e., Pb and I) and is relatively insen-

sitive to the light atoms (i.e., C, N, and H). A more detailed structural description of the geometrical configuration of the methylammonium groups was obtained by combining X-ray and neutron diffraction techniques.

The crystallographic data and refinement parameters of all the studied MAPI compounds are summarized in Table 1. Further details of the crystallographic data can be obtained from the CIFs (Crystallographic Information Files) deposited in the Inorganic Crystal Structure Database, CSD numbers: 2210263, 2210294, 2210328, 2210329, and 2210330.

**Table 1.** Crystallographic data, experimental details and structure refinement parameters of MAPI crystals synthesized in GBL and acid additives.

| Name | O | A | B | B$_{seed}$ | C |
|---|---|---|---|---|---|
| **Crystal data** | | | | | |
| Refined formula | PbI$_3$·24NC | PbI$_3$·0.5[N$_2$C$_2$] | PbI$_3$·NC | PbI$_3$·0.5[N$_2$C$_2$] | PbI$_3$·0.5[N$_2$C$_2$] |
| Temperature (K) | 296(2) | 299(2) | 298(2) | 301(2) | 296(2) |
| Crystal system | Cubic | Tetragonal | Orthorhombic | Orthorhombic | Tetragonal |
| Space group (no.) | *Pm-3m* (221) | *I4cm* (108) | *Fmmm* (69) | *Fmmm* (69) | *I4cm* (108) |
| *a* (Å) | 6.2955(6) | 8.9034(6) | 12.602(2) | 12.566(2) | 8.8972(6) |
| *b* (Å) | 6.2955(6) | 8.9034(6) | 12.602(2) | 12.594(4) | 8.8972(6) |
| *c* (Å) | 6.2955(6) | 12.6162(11) | 12.655(4) | 12.668(4) | 12.6214(9) |
| *V* (Å$^3$) | 249.51(7) | 1000.09(16) | 2009.7(8) | 2004.9(10) | 999.11(15) |
| *Z* | 1 | 4 | 8 | 8 | 4 |
| $\rho_{calc}$ (g/cm$^3$) | 4.086 | 4.077 | 4.058 | 4.068 | 4.081 |
| $\mu$ (mm$^{-1}$) | 26.099 | 26.050 | 25.922 | 25.985 | 26.072 |
| Radiation | Mo *K*$\alpha$ ($\lambda$ = 0.71073) | | | | |
| Crystal size (mm$^3$) | 0.030 × 0.040 × 0.040 | 0.040 × 0.050 × 0.080 | 0.030 × 0.040 × 0.050 | 0.020 × 0.020 × 0.020 | 0.035 × 0.050 × 0.090 |
| **Data collection** | | | | | |
| 2$\theta$ range (°) | 6.472–61.482 | 6.458–71.062 | 5.592–72.680 | 5.596–77.346 | 9.150–75.226 |
| F(000) | 254 | 1016 | 1939 | 2032 | 1116 |
| (sin $\theta$/$\lambda$)$_{max}$ (Å$^{-1}$) | 0.719 | 0.818 | 0.834 | 0.879 | 0.859 |
| *hkl* ranges | $-9 \leq h \leq 7$ $-8 \leq k \leq 9$ $-9 \leq l \leq 6$ | $-14 \leq h \leq 12$ $-14 \leq k \leq 12$ $-20 \leq l \leq 15$ | $-17 \leq h \leq 15$ $-20 \leq k \leq 20$ $-13 \leq l \leq 21$ | $-21 \leq h \leq 20$ $-22 \leq k \leq 22$ $-21 \leq l \leq 21$ | $-11 \leq h \leq 15$ $-15 \leq k \leq 15$ $-21 \leq l \leq 21$ |
| $T_{min} - T_{max}$ | 0.512–0.747 | 0.504–0.747 | 0.422–0.747 | 0.357–0.748 | 0.396–0.747 |
| Reflections collected/independent | 2632/111 | 5108/1114 | 4956/1188 | 7277/1550 | 9763/1330 |
| Reflections with $I/2\sigma(I)$ | 110 | 895 | 489 | 995 | 878 |
| $R_{int}$ | 0.0862 | 0.0356 | 0.0966 | 0.0520 | 0.0722 |
| $R_\sigma$ | 0.0196 | 0.0293 | 0.0999 | 0.0467 | 0.0445 |
| **Refinement** | | | | | |
| Data/restraints/parameters | 111/0/4 | 1114/1/26 | 1188/0/21 | 1550/0/22 | 1330/1/26 |
| Final *R* indexes [$F^2 > 2\sigma$ ($F^2$)] | $R_1$ = 0.0483, $R_w$ = 0.1269 | $R_1$ = 0.0682, $R_w$ = 0.1664 | $R_1$ = 0.1104, $R_w$ = 0.3340 | $R_1$ = 0.0411, $R_w$ = 0.0973 | $R_1$ = 0.0734, $R_w$ = 0.2310 |
| Final *R* indexes [all data] | $R_1$ = 0.0497, $R_w$ = 0.1274 | $R_1$ = 0.0826, $R_w$ = 0.1726 | $R_1$ = 0.1983, $R_w$ = 0.3979 | $R_1$ = 0.0758, $R_w$ = 0.1244 | $R_1$ = 0.1052, $R_w$ = 0.2585 |
| Goodness-of-fit on $F^2$ | 1.185 | 1.203 | 1.129 | 1.058 | 1.183 |
| Extinction coefficient | // | 0.0012(2) | 0.0010(3) | 0.00112(15) | 0.0069(14) |
| Largest diff. peak/hole (e·Å$^{-3}$) | +1.85/−3.28 | +3.75/−4.34 | +4.83/−8.49 | +2.10/−2.82 | +3.10/−2.67 |

The X-ray diffraction measurements were performed at room temperature (296–300 K). In the temperature range 165–327 K, MAPI has been reported mostly as tetragonal with different space groups: *I*4/*mcm* [30–32], *I*4*cm* [33,34], *I*422 [35], and *I*4/*m* [36]. In this study, single crystal X-ray diffraction data showed that the as-synthesized MAPI single crystals display various symmetries at room temperature (RT), which are discussed in detail hereafter.

## 3. Discussion

### 3.1. Crystal O: MAPI Crystallized without Additives

The *hkl* indexing procedure of the diffraction data of the MAPI crystals grown in the GBL, without acid additives, crystal O, revealed the following parameters: *a* = 6.280(7), *b* = 6.284(6) and *c* = 6.319(8) Å, $\alpha$ = 89.95(6), $\beta$ = 90.09(8) and $\gamma$ = 90.02(11)°. The collected reflections could also be indexed in the pseudo-cubic cell with *a* = 6.2955(6) Å. In the case of ambiguity related to cell choice, the general convention is to opt for the cell with higher symmetry. Furthermore, the accord among symmetry-equivalent reflections was taken into account. All the data collectively pointed to a cubic symmetry, and thus the structure of the MAPI obtained from the GBL solution was solved and refined in the cubic *Pm-3m* space group. Details of the relative refinement are given in Table 1.

Elsewhere [30,37] the authors asserted that the MAPI tetragonal phase solely existed in the temperature range $165 < T < 327$ K and underwent a tetragonal into cubic phase transition above 327 K. As a matter of fact, the occurrence of the cubic MAPI O sample (this work) at room temperature (RT) came as a major surprise as it was expected to be tetragonal.

Nonetheless, in more recent studies [38,39], transmission electron microscopy and X-ray diffraction on MAPI powder and film compounds clearly showed the coexistence of both the cubic and tetragonal phases at RT. The coexistence range was extended from 300 to 330 K, and the fraction of the tetragonal and cubic coexisting phases was estimated to be ~70/30% [40].

Kim and Uchida [41] reported that in the cooling process from the heated state to RT during the fabrication procedure of MAPI thin films (thickness: 300–500 nm), cubic-phase domains might survive in the synthesized compound.

In the case of MAPI crystal O, the cubic symmetry could be due to the crystallization conditions since the crystal's nucleation in GBL took place at a high temperature of 120 °C (393 K). Then the quenching from 120 °C might have led to large cubic-phase domains extended at the microscale in bulk.

In our refined model, Pb and I were positioned at $1a$ (0,0,0) and $3d$ (1/2,0,0), respectively, with a Pb–I interatomic distance of 3.1478 (Å). After the assignment of Pb and I, the highest electron residual peak (+3.58 e·A$^{-3}$) was observed at $1b$ (1/2,1/2,1/2). Figure 2 shows the projection of the cubic *Pm*-3*m* MAPI crystal structure down [100].

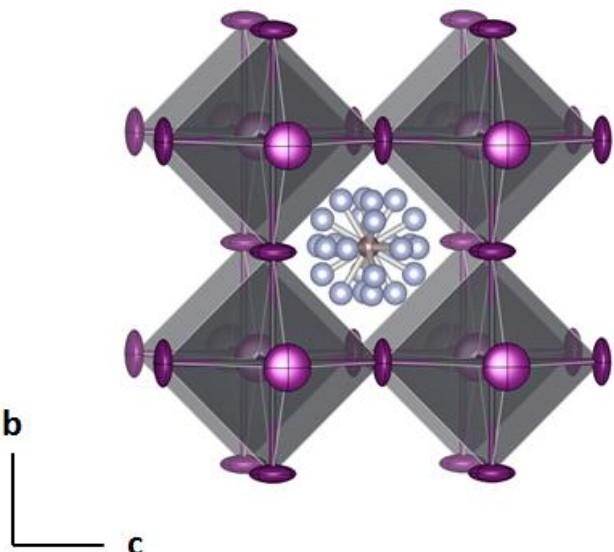

**Figure 2.** View of the cubic *Pm*-3*m* MAPI O (synthesized in γ-butyrolactone, GBL, without additives) crystal structure down [100]. PbI$_6$, grey octahedra; I, violet. The MA$^+$ (C: brown and N: blue) disorder is modelled over 24 equivalent positions inside the perovskite cage. Ellipsoids are shown with 50% probability.

By placing C on this site, we found 24 possible orientations, each in an occupancy of ~0.0417, for N at $24l$ (1/2,*y*,*z*) with a C–N bond length of 1.53 Å. This rotational disorder was consistent with the four-fold symmetry of the MA$^+$ cation described by Weller et al. [24] for CH$_3$NH$_3$PbI$_3$. The final refinement of the MAPI crystallized in GBL with the *Pm*-3*m* symmetry converged to $R_1 = 0.048$ and $wR_2 = 0.127$.

### 3.2. Crystal A: MAPI Crystallized with Acetic Acid as an Additive

The X-ray diffraction data of the MAPI-single crystals A indicated that the compound was tetragonal with cell parameters $a = b = 8.8958(6)$ Å and $c = 12.6320(9)$ Å (see Table 1) and values very similar to those previously reported [24,35,42]. The independent reflections

(1065 over 10,845 collected reflections) were consistent with both tetragonal Laue point groups at $4/m$ ($R_{int}$ = 0.070) and $4/mmm$ ($R_{int}$ = 0.072). The reflection conditions pointed to the body-centered "$I$" lattice. There were 581 reflections with $0kl$: $k + l = 2n$ that violated the $c$-glide, thus leading us to exclude the $I4/mcm$ space group commonly reported for $CH_3NH_3PbI_3$ [31,43,44].

The structure solution was initially performed in the $I4/mmm$ space group up to $R_1$~0.15. We then lowered the symmetry to $I4/m$ and then conducted a new refinement. Convergence was quickly achieved up to $R_1$ = 0.104 and $wR_2$ = 0.263. As already reported by Baikie et al. [37], the incompatibility of this symmetry with the perovskite hierarchy may result from the $PbI_6$ tilting disorder or the presence of twinning. In fact, when neglecting the systematic extinction violations of the $c$-glide, the structure of this compound could also be solved in the $I422$ ($R_1$ = 0.081 and $wR_2$ = 0.257), $I4cm$ ($R_1$ = 0.068 and $wR_2$ = 0.172) and in $I4/mcm$ ($R_1$ = 0.082, $wR_2$ = 0.242) space groups introducing the twinning laws: (-1/2-1/21/2 -1/2-1/2-1/2 1-10), (-100 0-10 00-1), and (-100 0-10 001), respectively. Actually, all the above-mentioned space group types have been documented in the literature. The MAPI structure is more often reported in one of the two tetragonal variants: the polar non-centrosymmetric $I4cm$ [45–47] and centrosymmetric $I4/mcm$ [48–50]. The subtle difference between these space groups, complicated by the presence of twin domains in the MAPI, makes it difficult to resolve the subtle structural details through the X-ray diffraction technique. It would be necessary to examine their electrical behavior under a magnetic field to reach a conclusion about the correct space group. If the MAPI exhibits ferroelectric properties, it has then $I4cm$ symmetry [51]. In this study, we chose to describe the final model of the MAPI grown using the mixture of GBL and acetic acid in $I4cm$ (refined as an inversion twin), with only 122/322 rejected reflections (over 5108 total observed) with $I > 3\sigma(I)$.

Figure 3 shows the tetragonal cell of crystal A projected down [-1-10] (a) and [1] (b). The atomic coordinates from our refinement (in $I4cm$) were in agreement with those reported by Stoumpos et al. [33] and Breternitz et al. [52].

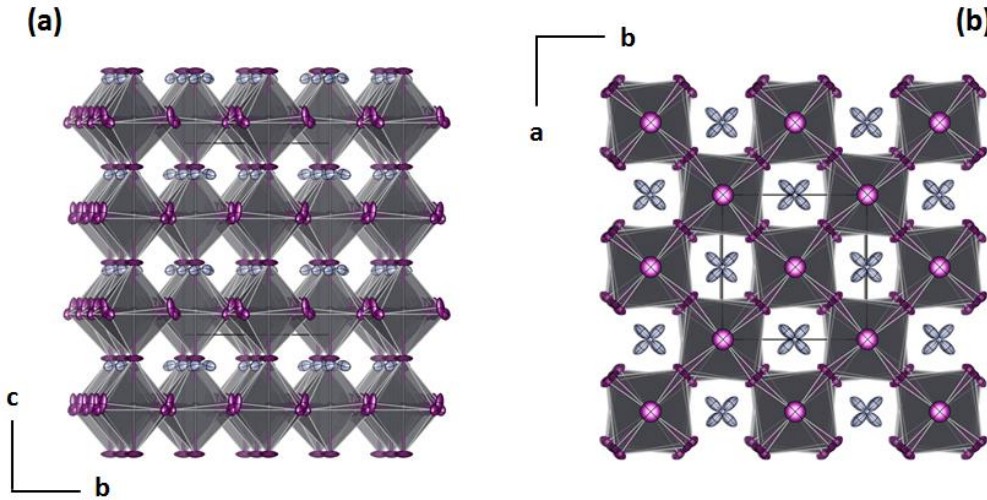

**Figure 3.** Projections of the MAPI "A" (synthesized in γ-butyrolactone, GBL, with acetic acid as an additive) crystal structure in the $I4cm$ symmetry: down [100] in (**a**), and down [1] in (**b**).

Pb, as well as I1, lies on general positions corresponding to the Wyckoff site $4a(0,0,z)$; I2 is disordered over two adjacent $8c$ Wyckoff sites. The C–N disorder has been refined using EXYZ and EADP constraints. The highest residual electron density peak (+3.75 e·Å$^{-3}$) was observed in the vicinity of I2. The interatomic distances observed for the $PbI_6$ octahedron (including the distorted I2 atom) are in the range from <3.135(15)–3.201(2) Å> with a mean Pb–I distance of 3.174 Å. The I–Pb–I angles are slightly distorted from the ideal 90° angle by about ±9° <81.7(3)–98.9(2)°> due to the disorder of I2. Those I2–Pb–I2 angles in the

*ab*-plane were observed in the range from <171.4(3)–172.1(12)°>. N was localized over the two equivalent sites, and 8*c* was statically distributed (50% occupancy).

### 3.3. Crystal B: MAPI Crystallized with Formic Acid as an Additive

The diffraction dataset indexing of the MAPI B grown in the mixture of GBL and formic acids, with FOR solution being three times (3×) concentrated, indicated the following cell dimensions: $a$ = 12.59, $b$ = 12.60, and $c$ = 12.63 Å (and $\alpha = \beta = \gamma = 90°$). The reflection conditions ($h + k$, $h + l$, $k + l = 2n$) were in agreement with the $F$ lattice. We first reduced the data in the cubic cell: $a$ = 12.632(17) Å. However, the structure refinement in the *Fm-3m* space group did not reach a good convergence ($R_1$ = 0.143).

Several trials in the tetragonal symmetry were conducted, but, at last, the structure of the MAPI B was solved and refined in the orthorhombic *Fmmm* space group, based on 1188 unique reflections ($R_{int}$ = 0.0966, $R_\sigma$ = 0.0999) out of over 4956 measured reflections, with $a = b$ = 12.602(2) Å, and slightly longer $c$ parameter $c$ = 12.655(4) Å. The orthorhombic *Fmmm* space group was already reported for the MAPI phase at RT in the work of Jaffe et al. [21].

In this structure, Pb lies on 8*e*(¼,¼,½) and bounds to I atoms on three independent positions: I1, I2, and I3 at 8*g*($x$,½,½), 8*h*(½,$y$,½), and 8*f*(¼,¼,¾), respectively. The N–C methylammonium disorder was refined over the 8*i*(0,½,$z$) site. The final refinement based on $F^2$, including the twin law (010 100 00-1), converged at $R_1$ = 0.11 and $wR_2$ = 0.397 for 489 reflections observed with $F_0 > 4\sigma(F_0)$. Despite the relatively high $R$ indices, the model provided a reliable basis for the interpretation of MAPI-B's crystal structure.

The Pb–I bonds range from 3.1573(7) to 3.1763(13) Å with an average bond length of 3.166 Å. The angles of the PbI$_6$ octahedra are somehow distorted and deviate about ±3.6° from the ideal 90° angle.

### 3.4. Crystal B$_{seed}$: MAPI Nucleated with Acetic Acid as an Additive and Grown with Formic Acid as an Additive

The crystal B$_{seed}$ crystallized in the orthorhombic system with cell dimensions: $a$ = 12.566(2), $b$ = 12.594(4), and $c$ = 12.668(4) Å. The reflection conditions ($h+k$, $k+l$, $h+l = 2n$) pointed to the $F$ lattice. A total of 7277 collected reflections yielded 1550 independent reflections in the *mmm* Laue group ($R_{int}$ = 0.052, $R_\sigma$ = 0.046).

The structure solution and the refinement were successfully achieved in the *Fmmm* space group ($R_1$ = 0.041 and $wR_2$ = 0.124) and, including the twin law (100 001 010), with a twin fraction of 36.8(5)%. The atomic coordinates of the Pb and I atoms in the structural model of the as-prepared MAPI slightly differ from that reported in the work of Jaffe et al. [21].

Pb is at 8*e* and links three independent I atoms: I1 was split over two equivalent neighboring Wyckoff sites, and 8*g* occupied at 50%; I2, 8*h*; and I3, 8*f*. The Pb–I octahedral distances, including the disordered iodide atom (I1), range from 3.1564(15) to 3.201(3) Å with an average bond length of 3.174 Å and the angles I–Pb–I are <86.1(3)–93.9(3)°> and <173.0(5)–180.0(5)°>. The geometry of the MA$^+$ cation was modelled by fixing the positions of N and C found at 16*m* and 16*n* in the *m*-symmetry sites, respectively. With this model (see Figure 4), the N and C resulted in two symmetrically equivalent positions inside the cubo-octahedral cavity (related through mirror planes perpendicular to the *a*- and *b*-axes), and each occupied 50% with a short C–N distance of 1.37 Å (this is slightly shorter compared with that reported by Kleber et al. [53]). Finally, the remaining electron residual densities ($\Delta\rho_{max}$ = +2.1 e·Å$^{-3}$) observed in this model were found at ~0.74 Å around I2.

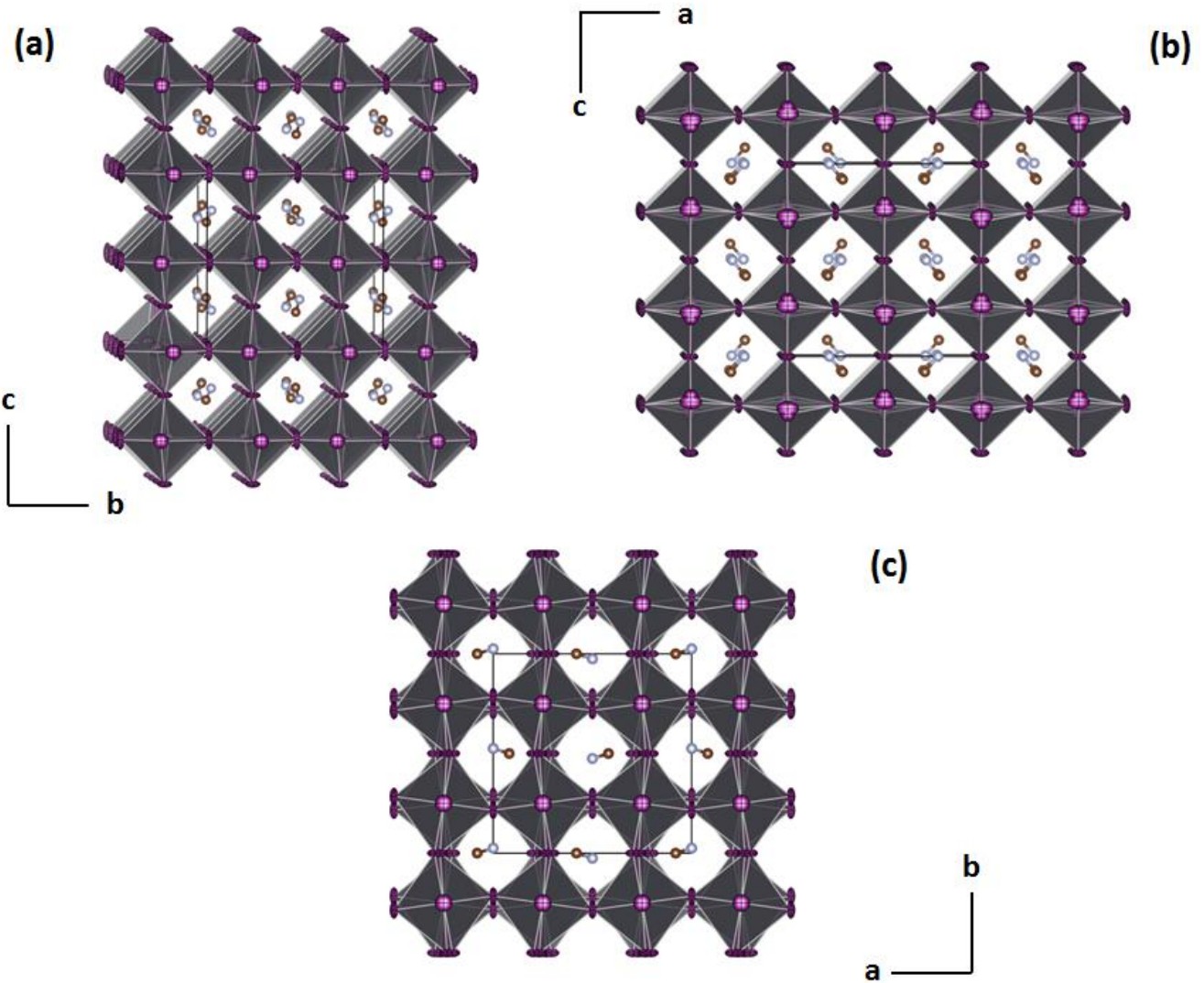

**Figure 4.** View of the crystal structure of MAPI B$_{seed}$ prepared from a seed obtained in γ-butyrolactone, GBL, with acetic acid as an additive and grown in a solution of formic acid in GBL. Projections of the orthorhombic *Fmmm* structure down [100] (**a**), down [10] (**b**), and down [1] (**c**).

### 3.5. Crystal C: MAPI Crystallized with Trifluoroacetic Acid as an Additive

The lattice parameters of crystal C were found to be: *a* = *b* = 8.8972(6) Å and *c* = 12.6214(9) Å. The crystal structure is tetragonal, lattice "*I*", and was solved in the *I4cm* space group ($R_1$ = 0.073, $wR_2$ = 0.259), introducing the inversion twinning (-100 0-10 00-1).

The atomic coordinates of all the atoms were placed as in the structural model of the MAPI grown in the mixture of GBL and acetic acid (crystal A). As in the structure of A, the iodide atom I2 displays positional disorder. It was refined by placing I2 over three sites: I2a, I2b and I2c and fixing their occupancies to 0.33, 0.33 and 0.34, respectively.

Both C and N were refined over two symmetrically equivalent sites that occupied 50% each. The remaining residual electron density peaks (+3.10 e·Å$^{-3}$) were observed at 0.94 Å from I2.

The PbI$_6$ are quite distorted, the mean Pb–I distance is 3.171 Å <3.1137(13)–3.205(6) Å> and bond angles I–Pb–I are in the ranges from <83.6(6)–97.3(6)> and <168.0(0)–180.0(0)>.

### 3.6. X-ray Powder Diffraction

The experimental X-ray powder diffraction (XRPD) pattern for sample B and sample B$_{seed}$ (intensity magnified 10×) are shown in Figure 5. The diffractograms of both samples

show the presence of characteristic peak splitting for $2\theta > 25°$ due to the symmetry lowering. The simulated XRPD patterns based on their single crystal CIFS clearly demonstrated the consistency of the experimental patterns with the *Fmmm* orthorhombic symmetry determined for MAPI crystallized in formic acid.

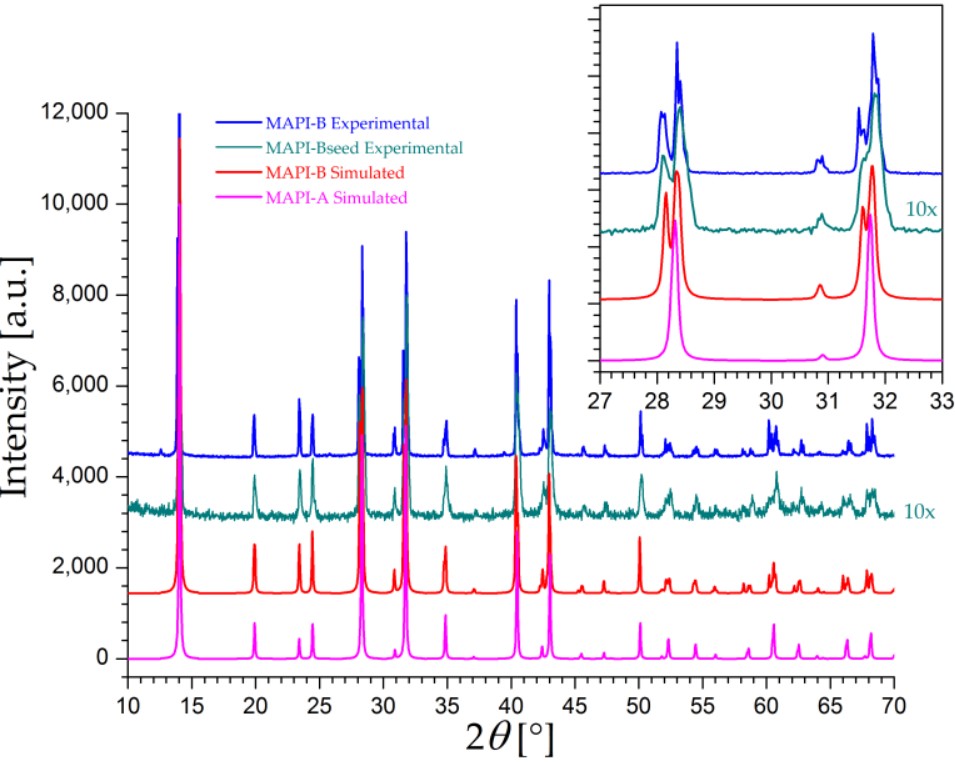

**Figure 5.** Experimental X-ray powder diffraction (XRPD) patterns for sample (blue) and sample B$_{seed}$ (cyan), its intensity was magnified 10×, and calculated XRD patterns of crystal B (red) and crystal A (magenta) based on structural refinement. The XRPD patterns of MAPI grown in formic acid are in agreement with the orthorhombic *Fmmm* symmetry.

### 3.7. Structural Comparison among the MAPI Crystals

All the synthesized MAPI crystals crystallized in a three-dimensional (3D) perovskite structure of the type ABX$_3$, with lead and iodide on the B and X sites. The 3D framework is built upon corner-sharing PbI$_6$ octahedra, which define cubo-octahedral cages where the MA$^+$ cation occupies the A-site.

A structural relationship exists between the symmetries reported in Table 2. The cell dimensions of the large tetragonal and orthorhombic cells are correlated to the small cubic one through symmetry transformation elements, as illustrated in Figure 6. The primitive cubic lattice *P* can be transformed into the tetragonal centered *I* cell by a 45° rotation of the *a*-parameter around the *c*-axis that changes the lattice constants in the manner $a_T = b_T = a_C \times \sqrt{2}$, and $c_T = 2 \times a_C$ (see Figure 6a), and into the orthorhombic lattice "*F*" by a translation of the cubic cell along the directions [100], [10], and [1] which brings them up to nearly equal lattice constants ($a_F \approx b_F \approx c_F \sim 2 \times a_C$) (see Figure 6b). The structural relationship between the tetragonal *I* and the orthorhombic *F* cells is shown in Figure 6c. The tetragonal cell was rotated around the diagonal [1-1-1] accompanied by a double translation along the [100] and [10] directions leading to the face-centered *Fmmm* orthorhombic cell with the following parameters: $a_F \approx a_T \times \sqrt{2}$, $b_F \approx b_T \times \sqrt{2}$, and $c_F \approx c_T$.

**Table 2.** Geometric parameters of the MAPI crystals from acid additives.

| Sample | Symmetry | Pb—I Bond Length (Å) | I—Pb—I Angle (°) |
|---|---|---|---|
| Crystal O | Cubic *Pm-3m* | 3.1478(3) | 90.0 180.0 |
| Crystal A | Tetragonal *I4cm* | <3.135(15)–3.201(2)> Avg. 3.174 | <81.7(3)–98.9(2)> <171.4(3)–180.0(0)> |
| Crystal B | Orthorhombic *Fmmm* | <3.1573(7)–3.1763(13)> Avg. 3.166 | <86.4(2)–93.6(2)> <172.5(3)–180.0(0)> |
| Crystal B$_{seed}$ | Orthorhombic *Fmmm* | <3.1564(15)–3.201(3)> Avg. 3.174 | <86.1(3)–93.9(3)> <173.0(3)–180.0(0)> |
| Crystal C | Tetragonal *I4cm* | <3.1137(13)–3.205(6)> Avg. 3.171 | <83.6(6)–97.3(6)> <168.0(0)–180.0(0)> |

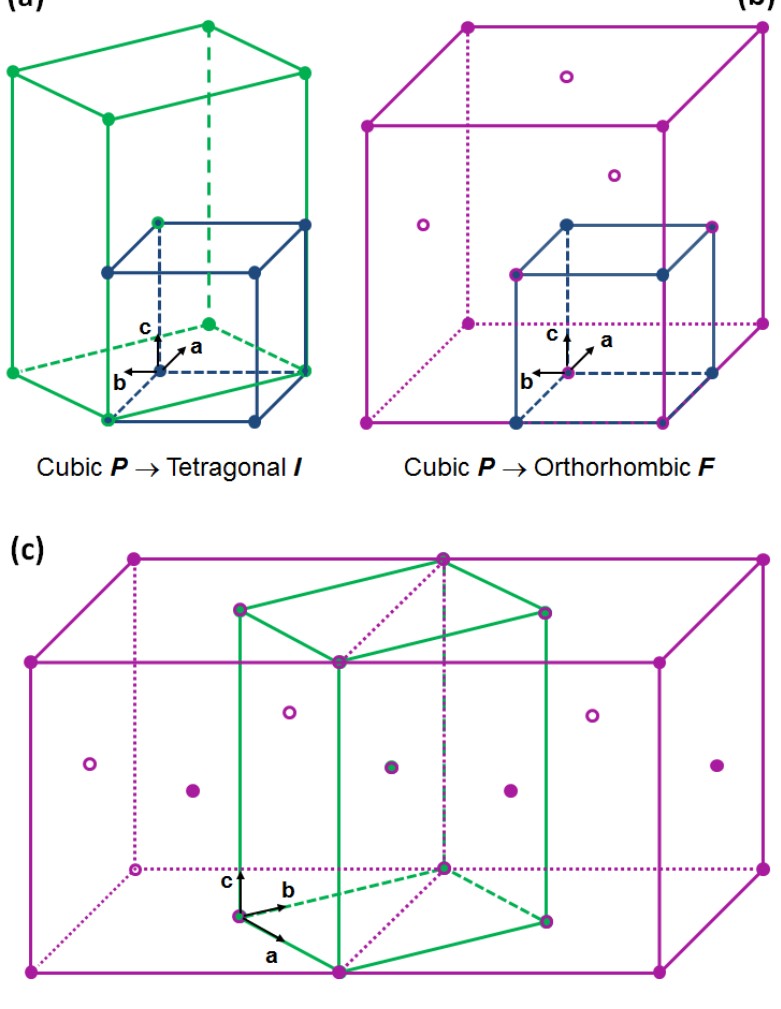

**Figure 6.** Schematic representation illustrating the symmetry relationship among the unit cells of the studied MAPI: (**a**) transformation from the cubic cell *P* to the tetragonal *I*; (**b**) from the cubic *P* to the orthorhombic *F*; and (**c**) from the tetragonal *I* to the orthorhombic *F*.

Taking into account the geometry relationship between the MAPI crystal structures and considering the intimate twinning, each MAPI reported herein was described in the symmetry that best fits its diffraction pattern.

Based on temperature-dependent [37,54–56] and pressure-induced [57,58] studies, it was suggested that the pphase transitions in the MAPI may originate from Pb–I contraction and elongation of I–Pb–I angle distortions coupled with octahedral $PbI_6$ tilting [37] and from the rotational behavior (dipole ordering caused by the H···I hydrogen bond) of $MA^+$ [55,59] in the perovskite cages. Since, here, MAPI crystals were studied by single-crystal X-ray diffraction at ambient conditions, the structural variation must be associated with the use of acid additives. In the cubic phase, the interatomic Pb—I distances are short (3.148 Å) and I–Pb–I angles are perfectly parallel to the crystal axes. This arrangement of the $PbI_6$ octahedra is described as $a^0a^0a^0$ following Glazer's notation [60], which describes the orientation of the octahedral tilting within the crystal directions in a given perovskite structure. On the other hand, the tetragonal and orthorhombic phases display longer and similar Pb–I bond lengths with an average value of 1.171 Å and distorted angles (see Table 2). In the tetragonal and orthorhombic phases, the perovskite structures show $PbI_6$ octahedral tilting of the type $a^0a^0c^-$.

Further work involving more sensitive diffraction techniques (e.g., neutrons or synchrotron radiation) is needed to analyze the structural dynamics of the MAPI perovskites obtained in this study.

### 3.8. Raman Characterization

Non-resonant Raman spectra on crystals A, B, $B_{seed}$, and C, obtained in GBL with short chains of carboxylic acids as additives, and on Crystal O obtained in GBL without additives, are shown in Figure 7. The spectra collected at room temperature (RT) do not show any differences and are all characterized by two energetic regions: mid-frequency features from 810–1770 $cm^{-1}$ and high-frequency features from 2770–3300 $cm^{-1}$. The distinctive Raman shifts denote vibrational modes characteristic of the tetragonal phase for all the investigated crystals [61].

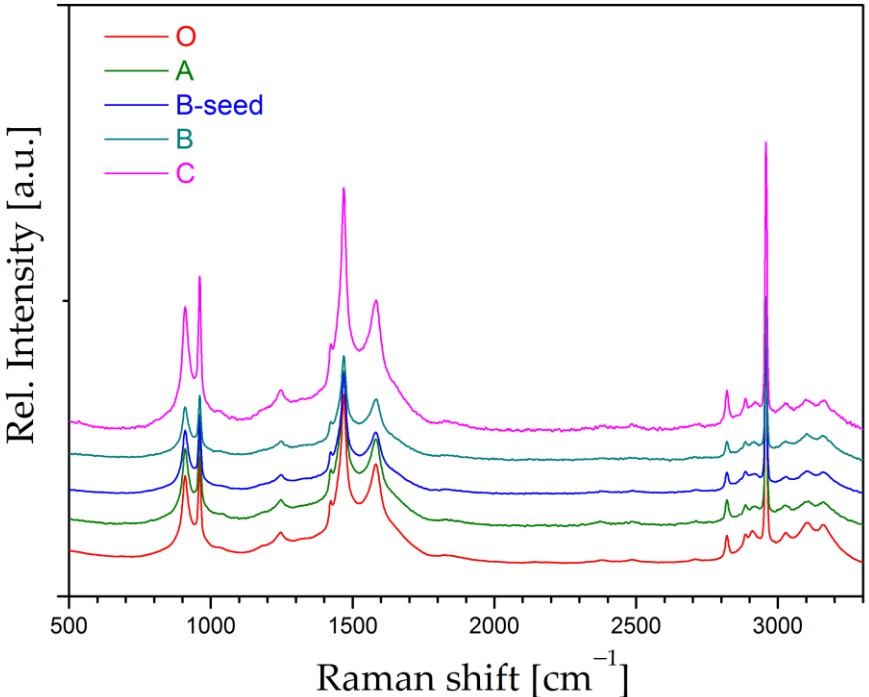

**Figure 7.** Normalized Raman spectra at room temperature (RT) of the hybrid perovskite crystals grown in γ-butyrolactone (GBL): crystal O, without additive; crystal A nucleated and grown with acetic acid; crystal B nucleated and grown with formic acid; crystal $B_{seed}$ nucleated with of acetic acid and grown in the presence of formic acid; crystal C nucleated and grown with trifluoroacetic acid.

In Poglitsch et al. [30], except for peak narrowing, no drastic change was observed in the Raman peaks of the cubic and tetragonal phases. The characteristic splitting ~907 cm$^{-1}$ occurred at 160 K, corresponding to the phase transition to the orthorhombic phase *Pna*2$_1$. More recent diffraction analysis reassigns the low-temperature orthorhombic phase to the *Pnma* space group [24]. It is noted to mention that PbI$_6$ octahedra in *Pna*2$_1$ and in *Pnma* are distorted, and the octahedral distortions are of the type $a^+b^-b^-$ and $a^-b^+a^-$, respectively, according to Glazer's notation. In our work, we found the *Fmmm* symmetry for B and B$_{seed}$, obtained with formic acid addition, in which the associated octahedral tilting $a^0a^0a^-$ is similar to the tetragonal *I4cm* symmetry. Furthermore, the MA$^+$ from the SC-XRD study is orientationally disordered in all the phases examined at room temperature. Due to the significant MA$^+$ disorder and the similarity in octahedral distortions of PbI$_6$ in the examined phases, no evident changes are detectable in the Raman spectra.

## 4. Materials and Methods

### 4.1. Synthesis of Hybrid Perovskite Crystals

The crystallization process was based on the seed-assisted inverse temperature crystallization (ITC) method. The crystallization solution was composed of the precursors: methylammonium iodide (CH$_3$NH$_3$I) and lead (II) iodide (PbI$_2$) dissolved in GBL at a concentration of 1 mol/L in a 1:1 molar ratio [13]. The solution was continuously stirred at 60 °C overnight and filtered through 0.2 μm PTFE filters to obtain a transparent MAPI solution. Three screw vials were filled, each with 2 mL of the MAPI solution. The vials were immersed in an oil bath at 110 °C for one hour before adding the acid additive. We selected three acid additives: acetic acid, formic acid, and trifluoroacetic acid. We added $5.3 \times 10^{-4}$ moles of each acid to the crystallization solution in each vial. In the case of the acetic acid and the trifluoroacetic acid, the acid addition induced a rapid generation of crystal seeds that were immediately extracted from the solution, dried, and stored in a nitrogen atmosphere. In the case of formic acid, a three times concentrated acid was needed to generate crystal seeds. Afterwards, a selected seed among the stored ones was immersed in a new MAPI solution at 110 °C. The seed was left at 110 °C to allow the growth of 3–5 millimeters in size. The crystal grown in the second step was filtered, rinsed in benzene, and dried under nitrogen.

Crystals of MAPI without any additives were obtained under the same conditions; nucleation was induced by initially increasing the temperature to 120 °C to generate the seed. Later, the seed was grown at 110 °C in a fresh MAPI solution to achieve a crystal 3–5 mm in size to be used as a reference. Details are reported in Table 3.

### 4.2. Acids' Effect on MAPI Crystallization

All the chosen acids (formic acid, acetic acid and trifluoroacetic acid) were miscible with the solvent GBL, and their pKa scale is inverse to the strength scale of the selected acids [62]. At the addition point of the acids, different effects on MAPI nucleation were observed depending on the state of the added acids.

Among the selected acids, only the acetic acid was in a liquid state at the chosen crystallization temperature, $T_c$ = 110 °C. Since the density of acetic acid (Table 3) is lower than the density of the solvent, GBL (D$_{GBL}$ = 1.13 g/mL), the addition of acetic acid instantly generated an interface with GBL that disappeared a few minutes after its spontaneous mixing with GBL. In these first instances, black crystallites formed at such an interface, which floated until they reached the weight that caused their precipitation on the bottom (crystal A).

The boiling temperature, $T_b$, of formic acid is below 110 °C. Hence, formic acid was in the gas state at $T_c$ and partially occupied the free volume overlying the crystallization solution. In these conditions, we found crystallites formed when the added moles of formic acid were at least three times more than the moles used for acetic acid [19] (crystal B).

**Table 3.** Details of the crystallization method of $CH_3NH_3PbI_3$ (MAPI) in GBL with acid additives.

| Crystal Name | Acid Name | $T_b$ § (°C) | Density at 25 °C (g/cm$^{-3}$) | Acid Constant (pKa) * | Volume of Added Acid (µL) |
|---|---|---|---|---|---|
| O | # | # | # | # | # |
| A | ACETIC | 118 | 1.05 | 4.74 | 30 |
| B | FORMIC | 100.8 | 1.22 | 3.75 | 57 |
| B$_{seed}$ | FORMIC | 100.8 | 1.22 | 3.75 | 19 |
| C | TRIFLUOROACETIC | 72.4 | 1.489 | −0.25 | 40 |

* Cologarithm of acid dissociation constant. § Boiling temperature.

The addition of the same moles of acid that generated crystallites for the acetic acid did not cause crystallites in the case of formic acid. It is worth noting that the same formic acid concentration could sustain the growth of a seed obtained with acetic acid as the additive (crystal B$_{seed}$).

Adding trifluoroacetic acid in the same concentration used for the acetic acid immediately generated crystallites that floated on the surface of the liquid phase until the biggest ones precipitated (crystal C).

Although formic and trifluoroacetic acids were both in the gas phase at $T_c$, they showed different effects on crystallite formation.

Since trifluoroacetic showed a more robust acidity than acetic and formic acids, the amount of trifluoroacetic acid dissolved in the liquid phase of the reagents resulted in a sufficient start for the crystallization immediately after the addition.

*4.3. Single Crystal X-ray Diffraction*

The crystals were cut into a reduced size suitable for the single crystal X-ray diffraction experiments and then mounted on MiTeGen sample mounts. The intensity data were collected at room temperature and ambient pressure conditions in a Bruker D8 Venture diffractometer equipped with an IµS 3.0 Incoatec microfocus source generating a monochromatic Mo $K_\alpha$ radiation ($\lambda$ = 0.71073 Å) and a Photon II CPAD detector. The set of diffraction data was collected in the $2\theta$ range 5.6–77.3° (in $\omega$ and $\varphi$ scans), reaching a completeness >96% and a resolution up to 0.5 Å$^{-1}$. SAINT and SADABS programs implemented in APEX 4 software were used to conduct intensity data integration and absorption correction [63]. Diffraction experiments were conducted on several specimens from each preparation, to verify the homogeneity of the crystals.

*4.4. X-ray Powder Diffraction*

X-ray powder diffraction (XRPD) patterns of samples B and B$_{seed}$ were collected on a Bruker D8 ADVANCE X-ray diffractometer working with a monochromated Cu $K\alpha$ radiation ($\lambda$ = 1.5406 Å) operating at 40 kV and 40 mA. The samples were laid on a Si flat plate holder. XRPD data were acquired at room temperature in ($2\theta/\theta$) Bragg-Brentano geometry in the angular range: $5 \leq 2\theta \leq 90°$ with a step size $\Delta 2\theta$ = 0.03° and exposure time of 1.5 s/step.

*4.5. Raman Spectroscopy*

Raman measurements were performed with a Bruker RFT 100 spectrometer with excitation with a 1064 nm Nd/YAG diode-pumped continuous wave (CW) laser. The laser spot on the sample was 1 mm in diameter. The spectra were collected in backscattered geometry with a liquid-nitrogen-cooled Ge diode detector, a 5 mm aperture, and a quartz beam splitter. The scan velocity was 5 kHz, the resolution was 4 cm$^{-1}$, and the laser power was kept at 200 mW. To ensure a good signal-to-noise ratio, more than 1064 scans were collected for each acid additive.

**5. Conclusions**

The results from comparing the effect of morphologically similar acids on MAPI crystallization evidenced the importance of the strength of the acid additive in MAPI crystallization. Experimental data confirmed the presence of an "acid threshold" to start the MAPI crystallization [17]. The presence of the short-chain acids in the crystallization ambient allowed the rapid growth of MAPI single crystals at 110 °C.

From the crystallographic investigation of MAPI crystals, which nucleated in the same conditions except for adding short-chain carboxylic acids, it was apparent that the acid additive affected the crystallization mechanism of these perovskites. We found that crystal O, prepared without additive, was cubic. Crystals B and B$_{seed}$, which were both grown with formic acid as an additive, crystallized as orthorhombic. Conversely, crystals A and C, prepared in acetic acid and trifluoroacetic acid, respectively, adopted a tetragonal symmetry. More sensitive techniques, such as neutron and synchrotron diffractions, are necessary in order to analyze in depth the structural dynamics of the obtained MAPI crystals.

Further study of other properties of MAPI single crystals obtained from short-chain carboxylic acids as additives is under way.

**Author Contributions:** Methodology and conceptualization, C.D. and M.G.; analysis, M.G. and G.R.; validation L.B.; writing—original draft preparation, C.D. and M.G.; supervision, L.B.; funding acquisition, L.B. and G.R. All authors have read and agreed to the published version of the manuscript.

**Funding:** This research was funded by Mission Innovative program MiSE under the Grant "Italian Materials Acceleration Platform-IEMAP". The financial support for the project "PRIN 2017, TEOREM–deciphering geological processes using Terrestrial and Extraterrestrial ORE Minerals (prot. 2017AK8C32)" allowed the performance of the diffraction measurements.

**Data Availability Statement:** The crystallographic data associated with this article are available in the Cambridge Crystallographic Data Center (CCDC). Crystal O, CCDC Number: 2210263. Crystal A, CCDC Number: 2210294. Crystal B, CCDC Number: 2210328. Crystal B$_{SEED}$, CCDC Number: 2210329. Crystal C, CCDC Number: 2210330.

**Acknowledgments:** C.D. and G.R. acknowledge funding from the Mission Innovation program MiSE under the Grant "Italian Energy Materials Acceleration Platform—IEMAP". M.G. and L.B. acknowledge CRIST, Centro di Studi per la Cristallografia Strutturale, University of Firenze, Firenze (Italy).

**Conflicts of Interest:** The authors declare no conflict of interest.

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
