# Peer review of "The Effect of Short Chain Carboxylic Acids as Additives on the Crystallization of Methylammonium Lead Triiodide (MAPI)"

_inorganics, doi:10.3390/inorganics10110201_

Round 1

Reviewer 1 Report

3D lead halide perovskites attracted a lot of attention in recent years due to their tunable optoelectronic properties. One of the most important representative is MAPbI3 beacuse this compound exhibits very high PCE of 25%. Since defects can significantly affect properties of these materials, it is important to study single crystals since they have much smaller number of defects compared to thin films or polycrystalline samples. Growth of large single crystals is often a challenging task and the method of growth can also affect morphology of the crystals and in some case stabilize different polymorphs. In the present manuscript, the authors report the effect of different additives on crystal growth and structure of MAPbI3. The most important conclusion is that different additives can stabilize different polymorphs. Although the subject is of great interest, I have serious doubts regarding this major conclusion and therefore more data are needed to support it. The secific comments are as follow:

1. Introduction is not well-written and should be improved. First of all, it is composed on many short paragraphs (15!), which should be combined into longer ones. I also miss some general information on 3D lead halide perovskites and growth methods. Therefore, a paragraph should be added at the beginning of the Introduction section, in which the authors should mention that 3D lead halide perovskites were discovered for only 4 organic cations: methylammonium, formamidinium, methylhydrazinium and aziridinium (citations should be added for each case). Then they should mention that these perovskites are of interest due to tunable optoelectronic properties, including NLO properties (in case of methylhydrazinium analogues), but iodides, especially MAPbI3 is of great interest due to its superior photovoltaic properties.  Then, the 4 first paragraphs in the present manuscript can be presented as one longer paragraph. After this, I recommed to add a new paragraph describing shortly different growth methods, at least for MAPbX3 crystals (anti-solvent method or method proposed by Fateev et al (10.1021/acs.chemmater.0c04060) can also lead to very large and good quality crystals). You can then add that one very promising method is ITC.

2. It is very surprising that the obtained crystals have different symmetries. Crystal symmetry of MAPbI3 should be tetragonal at RT if there are no doping with other organic cations like ethylammonium or dimethylammonium, which can stabilize cubic phase at RT (see some examples in literature both for MAPbI3 and MAPbBr3). However, it is very unlikely that the low-temperature orthorhombic phase would be stable at RT since this phase has ordered MA+ cations and due to small size, these cations can order completely only at low temperatures. All information regarding crystal structures is reported based on single-crystal X-ray diffraction. The authors should add powder X-ray diffractograms for all samples since they will very clearly show if the characteristic splitting due to lowering of cubic symmetry is still present or not. Furthermore, they should add simulated XRDpatterns based on their single-crystal cifs to see if the experimental patterns are consistent with the proposed symmetries.

3. The phase transition from the cubic to tetragonal phase is associated with a small DSC anomaly and that from the tetragonal to orthorphomic with very large DSC anomaly. Ttherefore, DSC data in the 300-120 K should be added: if the authors are right that some crystals have orthorhombic symmetry at RT, no strong DSC anomaly should be present near 162 K and for the cubic phase, a small anomaly should be present below 300 K due to transformation to the tetragonal phase.

4. As I mentioned, the orthorhombic phase should have ordered MA+ cations but the cubic one completely disordered MA+. This can be very clearly seen in the Raman spectra since for the disordered phase, the bands near 900 and 1600 cm-1 should be broad and weak but for the ordered orthorhombic phases these bands should be very narrow and strong (see for instance 10.3390/solids3010008 for MAPbBr3 analogue).  I recommend to add a figure with Raman spectra measured at RT for all samples to see if they show any indication of MA+ ordering due to lowering of crystal symmetry to the orthorhombic.

5. As far as I know, the correct space group for the tetragonal phase of MAPbI3 is I4/mcm. The authors decided to choose space group I4cm for crystals "A" and "C". Since this is noncentrosymmetric space group, it should show SHG signal. Therefore, please show that SHG signal is observed or revise the space groups to centrosymmetric. By the way, ferroelectricity or SHG-activity have not been supported in literature by any studies of MAPbI3 single crystals so it is very doubtful.

Author Response

Dear reviewer, we thank you for your comments.

We follow your indications to improve our manuscript. Please find hereafter in enclosed PDF file our answer.

Reviewer 2 Report

Dionigi and co.workers present the effects of acids additives on the growth of single crystals of MAPbI3 and their crystal structure. The study presented by the authors is interesting and has good relevance for tuning the crystal structures and electronic properties of emerging perovskite solar cells. The work presented by the authors may be publishable after making minor revisions to the manuscript.

1) In lines 166-168 the authors mention that crystal O is grown into cubic phase due to possible nucleation at higher temperatures. Did the authors try to nucleate the crystals at lower temperatures and check the effect of temperature on the crystal structure. Some more exposition is required in this regard.

2) The authors need to add more exposition on the structural transformations shown in Figure 5. Specifically, description of Figure 5B is ignored in the text.

3) The labelling of acid names in Table 4 is not appropriate. For example formic acid is labelled as acetic in third row and acetic acid is not labelled.

4) The structural transformation of tetragonal seed crystals grown in acetic acid to orthorhombic in the presence of formic acid vapors is interesting case. Did the authors try to do crystal growth of the hydrophobic seeds generated from trifluoroacetic acid (TFA)  in the presence of formic acid? It will be interesting to see how the surface fluoro groups of TFA nucleated crystals acts when further grown in formic acid environments.

Author Response

Dear reviewer, we thank you for your comments.

We follow your comments to improve our manuscript. Please find hereafter in enclosed PDF file our answer.

Reviewer 3 Report

In the present work, the authors have developed a novel and reproducible crystallization method for the selective generation of targeted MAPI crystals with a targeted symmetry using a series of acid additives. In general, the synthesis of perovskite single crystals with a pre-targeted symmetry is quite challenging and in this manuscript, the authors have described that by using a suitable acid additive during the crystallization process this challenge could be achieved. The development of pervoskite single crystals is an area of intense interest due to their greater optoelectronic properties and stability in photovoltaic applications compare to polycrystalline films. The present work will attract the attention of a broad range of audiences throughout the chemical community interested in photovoltaics. The experimental work has been competently performed with a high scientific standard, and the results have been scholarly presented. Therefore, I recommend the publication of this work in Inorganics in its current form pending these minor points:

Minor points:

1. Table 1, third-row ‘ACTEC’ should be replaced by‘ACETIC’

2. Throughout the manuscript, I missed the bulk purity and yield of all the reported single crystals. Did the authors also characterise the bulk material? The bulk purity of the material should be discussed in the manuscript.

Author Response

 We thank the reviewer for his comments. Please find our answer in the attached PDF file.

Round 2

Reviewer 1 Report

The authors revised the manuscript and provided clear evidence of the orthorhombic distortion for the samples grown from formic acid. I recommend acceptance of this paper.